# Hypoxaemia prevalence and management among children and adults presenting to primary care facilities in Uganda: A prospective cohort study

**Hamish R. Graham**[1,2]*, **Yewande Kamuntu**[3], **Jasmine Miller**[4], **Anna Barrett**[1,5], **Blasio Kunihira**[3], **Santa Engol**[3], **Lorraine Kabunga**[3], **Felix Lam**[4], **Charles Olaro**[6], **Harriet Ajilong**[7], **Freddy Eric Kitutu**[8,9]

**1** Melbourne Children's Global Health, MCRI, University of Melbourne, Royal Children's Hospital, Parkville, Victoria, Australia, **2** Department of Paediatrics, University College Hospital, Ibadan, Nigeria, **3** Clinton Health Access Initiative Uganda, Kampala, Uganda, **4** Clinton Health Access Initiative, Boston, MA, United States of America, **5** Nossal Institute of Global Health, University of Melbourne, Parkville, Australia, **6** Director Health Services, Office of the Director of Curative Services, Federal Ministry of Health, Kampala, Uganda, **7** Uganda Paediatric Association, Kampala, Uganda, **8** Department of Pharmacy, Makerere University School of Health Sciences, Kampala, Uganda, **9** Sustainable Pharmaceutical Systems (SPS) unit, Makerere University School of Health Sciences, Kampala, Uganda

* hamish.graham@rch.org.au

## Abstract

Hypoxaemia (low blood oxygen) is common among hospitalised patients, increasing the odds of death five-fold and requiring prompt detection and treatment. However, we know little about hypoxaemia prevalence in primary care and the role for pulse oximetry and oxygen therapy. This study assessed the prevalence and management of hypoxaemia at primary care facilities in Uganda. We conducted a cross sectional prevalence study and prospective cohort study of children with hypoxaemia in 30 primary care facilities in Uganda, Feb-Apr 2021. Clinical data collectors used handheld pulse oximeters to measure blood oxygen level (SpO$_2$) of all acutely unwell children, adolescents, and adults. We followed up a cohort of children aged under 15 years with SpO$_2$<93% by phone after 7 days to determine if the patient had attended another health facility, been admitted, or recovered. Primary outcome: proportion of children under 5 years of age with severe hypoxaemia (SpO$_2$<90%). Secondary outcomes: severe (SpO$_2$<90%) and moderate hypoxaemia (SpO$_2$ 90–93%) prevalence by age/sex/complaint; number of children with hypoxaemia referred, admitted and recovered. We included 1561 children U5, 935 children 5–14 years, and 3284 adolescents/adults 15+ years. Among children U5, the prevalence of severe hypoxaemia was 1.3% (95% CI 0.9 to 2.1); an additional 4.9% (3.9 to 6.1) had moderate hypoxaemia. Performing pulse oximetry according to World Health Organization guidelines exclusively on children with respiratory complaints would have missed 14% (3/21) of severe hypoxaemia and 11% (6/55) of moderate hypoxaemia. Hypoxaemia prevalence was low among children 5–14 years (0.3% severe, 1.1% moderate) and adolescents/adults 15+ years (0.1% severe, 0.5% moderate). A minority (12/27, 44%) of severely hypoxaemic patients were referred; 3 (12%) received oxygen. We followed 87 children aged under 15 years with SpO$_2$<93%, with

**Data Availability Statement:** All relevant data for this study is included in the manuscript and supplemental files.

**Funding:** This project was funded by a grant from the Bill and Melinda Gates Foundation (BMGF) INV-001132 and ELMA Philanthropies to the Clinton Health Access Initiative (CHAI). Any views or opinions presented are solely those of the author and do not necessarily represent those of BMGF or ELMA, unless otherwise specifically stated. The funders had no role in study design, data collection and analysis, decision to publish, or preparation of the manuscript.

**Competing interests:** I have read the journal's policy and the authors of this manuscript have the following competing interests: HG is an advisor to the Lifebox Foundation, UNICEF, and Unitaid on pulse oximetry; YK, JM, BK, SE, LK, FL are employed by CHAI who are implementing the oxygen program in Uganda; CO works for the Ministry of Health which provides funding and oversight for health facilities.

complete data for 61 (70%), finding low rates of referral (6/61, 10%), hospital attendance (10/61, 16%), and admission (6/61, 10%) with most (44/61, 72%) fully recovered at day 7. Barriers to referral included caregiver belief it was unnecessary (42/51, 82%), cost (8/51, 16%), and distance or lack of transport (3/51, 6%). Hypoxaemia is common among acutely unwell children under five years of age presenting to Ugandan primary care facilities. Routine pulse oximetry has potential to improve referral, management and clinical outcomes. Effectiveness, acceptability, and feasibility of pulse oximetry and oxygen therapy for primary care should be investigated in implementation trials, including economic analysis from health system and societal perspectives.

## Introduction

Low blood oxygen levels (hypoxaemia) are common among children and adults admitted to hospital and associated with a 5-fold increase in odds of death and thus requires prompt detection and treatment [1–4]. Pulse oximetry is a non-invasive and objective method of measuring blood oxygen levels which has greater sensitivity and specificity for hypoxaemia than clinical signs. Healthcare workers (HCWs) use bedside oximeters to detect hypoxaemia and guide patient care–particularly oxygen therapy and other respiratory support [5]. As such, pulse oximeters are a global standard of care for hypoxaemia management and are regarded as a "priority medical device" by WHO [6–8].

Current WHO primary care guidelines recommend pulse oximetry "if available" for children U5 with respiratory complaints, with recommendation for referral if $SpO_2<90\%$ [9]. However, pulse oximetry availability and use in primary care settings is rare [10–12]. Previous studies in Africa and Asia-Pacific regions have shown that very few HCWs in primary care facilities have pulse oximeters or are trained to use them, even though it is a simple and acceptable technology [10–15]. The COVID19 pandemic has brought overdue public urgency and awareness of the longstanding need to strengthen pulse oximetry and oxygen systems in healthcare facilities in low- and middle-income countries (LMICs)–but we have limited data to guide pulse oximetry and oxygen therapy implementation in primary care settings [16–22].

Limited data suggests that pulse oximetry performed by primary care healthcare workers (HCWs) is feasible and can improve clinical decision-making and referral outcomes [11,14]. Even in settings where oxygen therapy may not be available, pulse oximetry remains useful as it enables HCWs to identify and prioritise severely ill patients for immediate care and referral to more equipped or specialized healthcare facilities where appropriate. In addition to pulse oximetry's role as a diagnostic tool (identifying patients who warrant oxygen therapy) and prognostic tool (identifying severely ill patients at high risk of death) [23], pulse oximetry can be a communication tool (facilitating information exchange between HCWs and caregivers/patients) [24].

Previous modelling estimates have suggested that increased availability and use of pulse oximetry, accompanied by reliable oxygen delivery systems, could save 148,000 child pneumonia deaths annually in 15 high-mortality countries [25]. However, these estimates are based on very limited data, including very poor data on the prevalence of hypoxemia in primary care settings. We are aware of only four studies that report hypoxaemia prevalence in primary care settings, finding that 5–15% of children presenting with pneumonia in the Gambia, Malawi and Ethiopia had $SpO_2<90\%$ and 1.4% of children presenting with any acute illness in Papua New Guinea had $SpO_2\leq90\%$ [11,14,26,27]. While these prevalence estimates are lower than

those in hospital settings (~15–30%) [1,3,4], they suggest a large number of severely ill patients with hypoxaemia presenting to primary care and not attending hospital.

Better understanding of hypoxaemia prevalence and current care practices in primary care will enable informed discussion about who should be screened with pulse oximetry (e.g. all patients versus just those with respiratory complaints), where it will be most feasible and cost-effective (e.g. pulse oximetry may not be warranted in small facilities that see very few hypoxaemic patients), and what effects can be expected from introducing pulse oximetry (e.g. numbers of patient referred and burden on larger hospitals) [10].

This study aimed to assess the prevalence and management of hypoxaemia among children and adults presenting to level III health centres (HCIII) facilities in Busoga and North Buganda regions in Uganda.

## Materials and methods

### Study design

We conducted a cross-sectional assessment of hypoxaemia among children, adolescents and adults, and prospectively followed a cohort of children with hypoxaemia to assess management and outcomes. The study was conducted between February and April 2021 at HCIIIs in low-altitude (1100–1400 meters) Busoga and North Buganda regions of Uganda that represent the catchment areas of Mubende and Jinja regional referral hospitals (S1 Fig).

### Study setting

This study was conducted as part of a partnership between the Ugandan Ministry of Health (MOH) and Clinton Health Access Initiative (CHAI) to improve access to pulse oximetry and oxygen therapy in Uganda. Uganda, a landlocked country in east Africa with 44 million population, has made substantial gains in economic and health development over recent decades [28]. The leading causes of death in Uganda remain preventable infectious diseases, including pneumonia, malaria, and HIV/AIDS, and neonatal conditions [28].

The Ugandan MOH has invested strongly in hospital oxygen services, guided by the National Scale-up of Medical Oxygen Implementation Plan [29]. The plan describes a clear role for pulse oximetry and oxygen therapy in hospitals, but gives little guidance on the role for pulse oximetry and oxygen therapy at lower level health facilities [29].

The Uganda health system is organized in tiered structure with five levels and designated service and treatment capacities (Table 1) [30,31]. Uganda has approximately 956 HCIIIs nationally, providing core preventive and curative outpatient services and maternity services to the local population [30]. In general, patients identified at HCIII (or HCII) as requiring overnight admission are referred to HCIVs or hospitals [29]. This study focused on government HCIII facilities as they provide essential frontline health services but are not currently included in plans for pulse oximetry or oxygen service scale up.

### Selection of healthcare facilities

This study was conducted in 5 districts in North Buganda region and 11 districts in Busoga region, representing the catchment areas of Mubende and Jinja Regional Referral Hospitals.

Study healthcare facilities were arrived at through a multi-stage and purposive sampling approach. We identified 120 HCIIIs in the target districts (80 in Busoga, 40 in North Central/Buganda), ranging between four and 15 per district. To obtain a representative sample, we entered all the facilities into an Excel spreadsheet (Microsoft Corp, Redmond, WA, US), used the *RAND* function to generate random numbers, sorted by size of random number, and

Table 1. Overview of the levels of health facilities in Uganda, and typical health services provided [30].

| Health facility level | Typical services provided |
| --- | --- |
| National Referral Hospital | Comprehensive specialist health services. Health research and teaching. Located in capital. Number of facilities nationally = 1 |
| Regional Referral Hospital | Specialist health services (e.g., psychiatry, ophthalmology, ENT surgery), and all general services–inpatient and outpatient. Health research and teaching. Cover multiple districts. N = 14 |
| District General Hospital | Preventive, promotive, curative maternity, paediatric, medical, surgical, laboratory and medical imaging services–inpatient and outpatient. In-service training, limited research. N = 52 |
| Health Centre IV | Preventive, promotive, and curative care, including maternity, in-patient and emergency surgical services. Serves a county. N = 180 |
| Health Centre III | Preventive, promotive, and curative care through out-patient services. Supervises HCII and community outreach activities. N~956 |
| Health Centre II | Basic preventive, promotive and curative care through out-patient and community outreach activities. |
| Village Health Teams | Facilitate health promotion, service delivery, community participation, access, and utilization of health services, through community outreach. |

selected a pre-specified number of facilities from each district based on each district's population (proportionate stratified sampling). We chose a total of 30 facilities (25% of all facilities) to ensure all districts contributed at least one facility. BK conducted randomisation and selection under supervision (YK, JM) and shared it with the broader team for actioning. After initial randomisation and selection of 30 facilities, we excluded 3 HCIIIs due to inaccessibility (2 located on an island, 1 located inside an army barracks), replacing them with other HCIIIs in their respective district according to the original random number allocation. While HCIIIs and their patient population vary across Uganda, we expected this to be broadly representative of patients presenting to HCIIIs in Uganda [32].

## Sample size and selection of study population

We considered all patients presenting to the HCIII with an acute illness to be eligible for inclusion, expecting most to be children under 5 years of age (U5). We excluded patients visiting the facility for routine pregnancy care, immunisation, child growth monitoring, or other non-acute health education or counselling activities. We aimed to capture all eligible patients presenting to participating facilities during the data collection period.

Extrapolating from limited available data we expected around 1% of children under five years of age to have severe hypoxaemia ($SpO_2$<90%) [1,14,26], and estimated that we would need 3,500 children U5 to estimate the point-prevalence of severe hypoxaemia with a margin of error of 0.5%. Based on admission data from participating facilities, this could be achieved with 70 HCIII-weeks of observation assuming low refusal to consent rate (10%).

We followed up a cohort of children with moderate or severe hypoxaemia ($SpO_2$<93% aged <15 years) to identify gaps in hypoxaemia management and opportunities for pulse oximetry to improve referral outcomes for this at-risk population.

## Data collection

Trained clinical data collectors were located at each included facility for 4 weeks during operating hours, and were responsible for identification of eligible participants, individual patient/caregiver informed consent, and survey completion. In practice, this typically meant data collectors were located at the outpatient clinic registration area, approached all patients/caregivers on arrival, conducted verbal informed consent if eligible, then immediately completed the

patient interview and measurement of blood oxygen saturation using pulse oximetry ($SpO_2$). Data collectors then met the patient after they had seen the healthcare worker to record diagnosis and treatment recommendations.

Data collectors used a standardised electronic data collection form to collect demographic and clinical data, including details of presenting complaint, signs/symptoms, healthcare worker diagnosis, and treatment recommendations. If $SpO_2$ was less than 93% and the patient was aged under 15 years, patients/caregivers were also invited to consent to a follow up phone call after 7 days. For this cohort, data collectors called the recorded phone number 7 days after presentation (repeated on 3 separate days if no response) to determine if the patient had attended another health facility, been admitted, or recovered. Data collection forms were pilot tested and revised with input from the local team and academic partners prior to formal data collection commencing. BK and YK supervised data collection activities and JM reviewed interim data for completeness and consistency. All communication with participants was in the respective local language (Lusoga in Busoga region, Luganda in Buganda region). Informed consent was conducted verbally with patients 15 years and above or caregivers of younger children and recorded electronically.

Data collectors used handheld pulse oximeters (Biotech Health Care Systems, Indore, India; Edan, San Diego, US) to obtain $SpO_2$ readings, preferentially applying the probe to the toe or finger and attempting readings three times before giving up. Recorded $SpO_2$ readings were documented alongside other vital signs at triage and provided to the treating healthcare worker with recommendation for referral for those with $SpO_2$ less than 90%. Project staff provided practical assistance for referral of patients when necessary, however, final decision lay with the treating healthcare worker and patient/family.

## Outcomes

Our primary outcome was the proportion of acutely unwell children U5 presenting to HCIIIs with severe hypoxaemia ($SpO_2$ <90%) [33].

We defined severe hypoxaemia as $SpO_2$ <90%, indicating a severely unwell cohort warranting oxygen therapy and admission to hospital according to WHO guidelines [9,33,34]. We defined moderate hypoxaemia as $SpO_2$ 90–93%, indicating an unwell cohort with compromised oxygenation warranting consideration for referral, admission and oxygen therapy, particularly for those with additional risk factors (e.g. severe anaemia, heart failure, acute neurological condition, malnutrition) [33,34].

Secondary outcomes included: moderate ($SpO_2$ 90–93%) and severe hypoxaemia ($SpO_2$ <90%) prevalence by presenting complaint, healthcare worker diagnosis, and age; proportion of hypoxaemic children U15 who were referred, attended hospital, were admitted, and recovered (versus still unwell/died). Referral was based on HCW record while hospital attendance, admission, and recovery were based on caregiver report on phone follow up.

## Statistical analysis

We reported hypoxaemia prevalence descriptively by age group and presenting complaint, using point prevalence with 95% confidence intervals and standard tests of difference in proportions (chi-squared or Fisher's exact as appropriate) to compare groups. We classified presenting complaints based on patient/caregiver reported symptoms and diagnosis based on HCW classification. We explored predictors of hypoxaemia using mixed effects logistic regression, including presenting complaint, age and sex as fixed effects and adjusting for clustering at the facility level by including facility as a random effect. We used backwards stepwise selection to identify independent predictive variables for hypoxaemia in each of the major age

categories (U5, U15, 5–14 years, 15+ years), using P-values as the primary decision point for dropping variables at each step.

We reported summary statistics on referral and oxygen therapy by age, as recorded by HCWs. We explored predictors of referral by age group using a mixed-effects model, adjusting for hypoxaemia ($SpO_2 < 94\%$), presenting complaint, age, and sex, and clustering at the facility level.

We mapped the pathway of care, reporting the proportion of children U15 with $SpO_2 < 93\%$ who were referred, attended hospital, were admitted, and recovered (versus still unwell/died).

## Approvals

We obtained ethical approval for this study from Makerere University School of Health Sciences Research and Ethics Committee (MakSHS REC #SHSREC Ref No.: 2020–030), and it was cleared and registered by the Uganda National Council of Science and Technology, reference number (HS631ES).

## Patient and public involvement

This study was developed and conducted in partnership with representatives from the Ugandan Ministry of Health (MoH), Makerere University and Ugandan Paediatric Association, who participated in a technical review group alongside study investigators. Patients, families, or civil society organisations were not involved in the study activities, except in relation to informed consent and direct participation.

## Results

We approached 7036 potentially eligible participants and successfully enrolled 5813 (83%). The major reason for exclusion were children who attended without an adult caregiver for consent (n = 1143). We recorded blood oxygen saturation using pulse oximetry ($SpO_2$) on 5788 (99.6%) participants, including 1561/1575 (99.1%) children U5 (**Fig 1**).

### Characteristics of participants

Participants represented all age ranges from newborns to older age (**S1 Table**). Among 1561 children under 5 years, the most reported presenting complaints were fever (77%), respiratory (74%), and diarrhoea/vomiting (29%). The most common HCW diagnoses were malaria (53%), acute respiratory infection (ARI, 52%), and diarrhoeal disease (16%). The leading presenting complaints and diagnoses among 935 older children aged 5 to 14 years were similar to younger children, with greater preponderance of non-abdominal pain (48%).

Among those over 15 years of age, the most reported presenting complaints were pain (66%), fever (41%), abdominal (38%), and respiratory (37%) (**S1 Table**). The most common HCW diagnoses were ARI (31%), malaria (29%), urinary or genital tract infection (19%), or non-diarrhoeal gastrointestinal disease (11%). Participants in Busoga tended to be younger (median 19 versus 22 years) and more female (71% versus 61%) compared to those in North Buganda.

### Prevalence of hypoxaemia

Overall hypoxaemia ($SpO_2 < 94\%$) prevalence among children and adults presenting with acute illness to HCIII primary care facilities in Uganda was 2.2% (95% CI 1.9 to 2.7) (**Table 2**). Hypoxaemia burden was greatest among infants under 1 year of age (39/392, 10%) and children aged 1–4 years (58/1169, 5%) who together accounted for 75% of hypoxaemia cases.

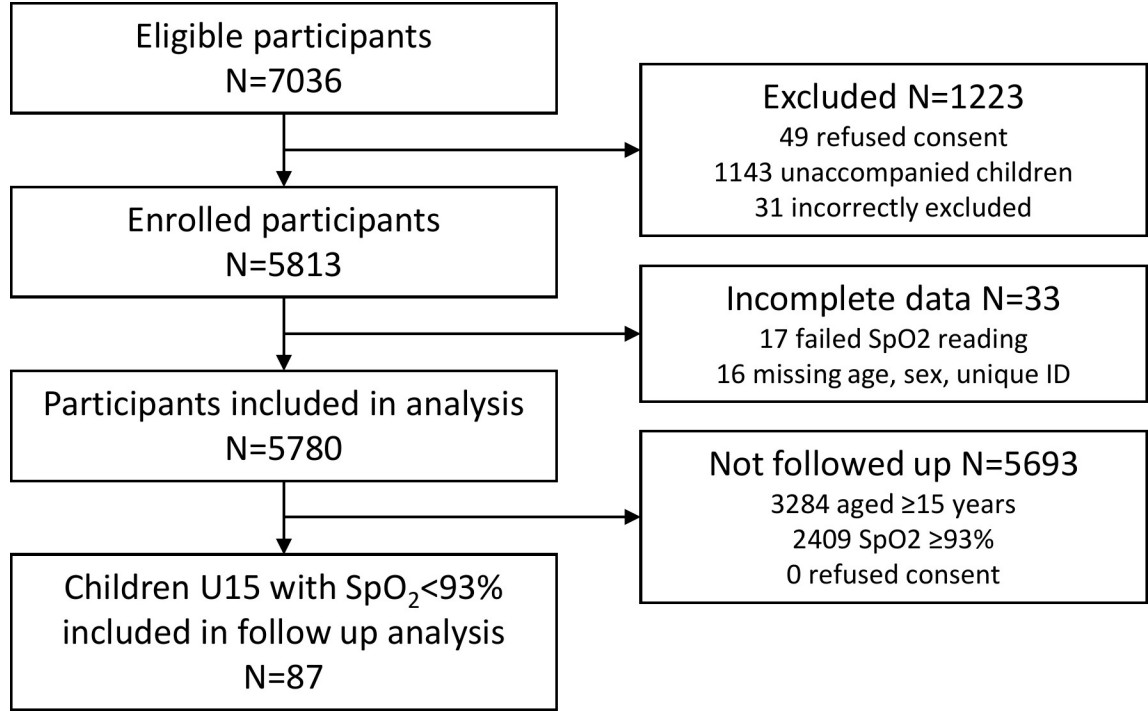

**Fig 1. Participant flow chart.**

Hypoxaemia prevalence showed a bimodal distribution with age, increasing again in older adulthood (**S2 Fig**).

**Children aged <5 years.** Among children U5, the prevalence of severe hypoxaemia (SpO$_2$<90%) was 1.3% (95% CI 0.9 to 2.1) and 4.9% (3.9 to 6.1) had moderate hypoxaemia (SpO$_2$ 90–93%) (**Table 2**).

Hypoxaemia (SpO$_2$<94%) prevalence was highest among those presenting with respiratory complaints (7.3%, 6.0 to 9.0) or fever (6.1%, 4.9 to 7.6) (**S2 Table**). Hypoxaemia was uncommon among those without fever or respiratory complaints (1/83, 1.2%). Performing pulse oximetry according to WHO IMCI guidelines exclusively on children U5 with respiratory tract complaints would have captured 86% (18/21) of severe hypoxaemia (SpO$_2$<90%) and 88% (67/76, 88%) of moderate hypoxaemia (SpO$_2$ 90–93%) among children U5.

Hypoxaemia (SpO$_2$<94%) prevalence was most common among children diagnosed with ARI (8.1%, 6.4 to 10.2), malaria (5.3%, 4.0 to 7.1), or sepsis (17.1%, 8.2 to 32.3), and particularly high for those diagnosed with pneumonia (18.8%, 13.0 to 26.4) (**Fig 2**, **Table 2**).

**Children aged 5–14 years.** Among children aged 5–14 years, the prevalence of severe hypoxaemia (SpO$_2$<90%) was 0.3% (0.1 to 1.0) and 1.1% (0.6 to 2.0) had moderate hypoxaemia (SpO$_2$ 90–93%) (**S2 Table**). Hypoxaemia (SpO$_2$<94%) was most common among older children presenting with respiratory symptoms (2.2%), fevers (1.8%), or pain (1.6%). ARI and malaria accounted for 85% (11/13) of hypoxaemia cases.

**Adolescents and adults ≥15 years.** Compared to children, hypoxaemia prevalence among adolescents and adults aged over 15 years was lower and less predicted by presenting complaint or diagnosis (**S2 Table**). While, hypoxaemia (SpO$_2$<94%) was most common among those presenting with respiratory complaints (0.8%, 0.4 to 1.4) or fever (0.5%, 0.3 to 1.1), one-third (7/20, 35%) of adolescents and adults with hypoxaemia did not have either fever or respiratory complaints.

**Table 2. Prevalence of moderate (SpO$_2$ 90–93%) and severe hypoxaemia (SpO$_2$<90%) among acutely unwell children, adolescents and adults presenting to primary care (HCIII) facilities in Uganda, Feb-Apr 2021.**

| | Total | Severe hypoxaemia (SpO$_2$<90%) | | | | Moderate hypoxaemia (SpO$_2$ 90–93%) | | | |
|---|---|---|---|---|---|---|---|---|---|
| | | N | prevalence | 95% CI | | N | prevalence | 95% CI | |
| **Neonate** | 16 | 0 | . | . | . | 4 | 25.0% | 8.9% | 53.3% |
| **1–11 months** | 376 | 8 | 2.1% | 1.1% | 4.2% | 27 | 7.2% | 5.0% | 10.3% |
| **1–4 years** | 1,169 | 13 | 1.1% | 0.6% | 1.9% | 45 | 3.8% | 2.9% | 5.1% |
| **5–9 years** | 633 | 3 | 0.5% | 0.2% | 1.5% | 5 | 0.8% | 0.3% | 1.9% |
| **10–14 years** | 302 | 0 | . | . | . | 5 | 1.7% | 0.7% | 3.9% |
| **15–24 years** | 1,088 | 0 | . | . | . | 3 | 0.3% | 0.1% | 0.9% |
| **25–49 years** | 1,602 | 1 | 0.1% | 0.0% | 0.4% | 7 | 0.4% | 0.2% | 0.9% |
| **50+ years** | 594 | 2 | 0.3% | 0.1% | 1.3% | 7 | 1.2% | 0.6% | 2.5% |
| *Total* | **5780** | **27** | **0.5%** | **0.3%** | **0.7%** | **103** | **1.8%** | **1.5%** | **2.2%** |
| **Girls (<5 years)** | 790 | 14 | 1.8% | 1.1% | 3.0% | 38 | 4.8% | 3.5% | 6.5% |
| **Boys (<5 years)** | 771 | 7 | 0.9% | 0.4% | 1.9% | 38 | 4.9% | 3.6% | 6.7% |
| **Girls (5–14 years)** | 568 | 1 | 0.2% | 0.0% | 1.2% | 4 | 0.7% | 0.3% | 1.9% |
| **Boys (5–14 years)** | 367 | 2 | 0.5% | 0.1% | 2.2% | 6 | 1.6% | 0.7% | 3.6% |
| **Women (≥15 years)** | 2,561 | 1 | 0.0% | 0.0% | 0.3% | 14 | 0.5% | 0.3% | 0.9% |
| **Men (≥15 years)** | 723 | 2 | 0.3% | 0.1% | 1.1% | 3 | 0.4% | 0.1% | 1.3% |
| **Busoga** | 4,120 | 20 | 0.5% | 0.3% | 0.7% | 86 | 2.1% | 1.7% | 2.6% |
| **North Central** | 1,660 | 7 | 0.4% | 0.2% | 0.9% | 17 | 1.0% | 0.6% | 1.6% |
| **U5 YEARS BY PRESENTING COMPLAINT** | | | | | | | | | |
| **Resp OR fever** | 1,478 | 21 | 1.4% | 0.9% | 2.2% | 75 | 5.1% | 4.1% | 6.3% |
| **Respiratory** | 1,158 | 18 | 1.6% | 1.0% | 2.5% | 67 | 5.8% | 4.6% | 7.3% |
| **Fever** | 1,208 | 15 | 1.2% | 0.7% | 2.1% | 59 | 4.9% | 3.8% | 6.3% |
| **Diarrhoea/Vomit** | 459 | 5 | 1.1% | 0.5% | 2.6% | 17 | 3.7% | 2.3% | 5.9% |
| **Malaria** | 825 | 10 | 1.2% | 0.7% | 2.2% | 34 | 4.1% | 3.0% | 5.7% |
| **ARI** | 811 | 13 | 1.6% | 0.9% | 2.7% | 53 | 6.5% | 5.0% | 8.5% |
| **"Pneumonia"** | 133 | 7 | 5.3% | 2.5% | 10.7% | 18 | 13.5% | 8.7% | 20.5% |
| **Diarrhoeal disease** | 243 | 3 | 1.2% | 0.4% | 3.8% | 10 | 4.1% | 2.2% | 7.5% |
| **Sepsis** | 41 | 0 | . | . | . | 7 | 17.1% | 8.2% | 32.3% |
| *Total U5* | **1,561** | **21** | **1.3%** | **0.9%** | **2.1%** | **76** | **4.9%** | **3.9%** | **6.1%** |

Notes: U5 = under 5 years of age; presenting complaints reported by patient or caregiver; clinician diagnosis recorded by treating healthcare worker; ARI = acute respiratory infection; Sepsis = bacteraemia, septic wound.

## Predictors of hypoxaemia (SpO$_2$<94%)

Among children U5 years, adjusted analysis identified hypoxaemia (SpO$_2$<94%) to be associated with age (aOR 0.66, 95% CI 0.54 to 0.79) and the presence of respiratory complaints (aOR 2.46, 1.30–4.66) with similar results for those 5–14 years (**S3 Table**). For those 15 years and over, age was the only variable independently associated with hypoxaemia, with older age associated with higher risk of hypoxaemia (aOR 1.05, 1.02–1.07).

## Referral and management

Overall, 32 (0.55%) participants were referred by HCWs to a higher-level facility, with minimal variation across age groups but some variation between North Central and Busoga region (1.1% versus 0.3%, P = 0.001). While one-third of those referred (12/32, 38%) had SpO$_2$<90%, and hypoxaemia (SpO$_2$<94%) was the strongest predictor of referral (**S4 Table**), less than half of those with severe hypoxaemia (SpO$_2$<90%) were referred (12/27, 44%).

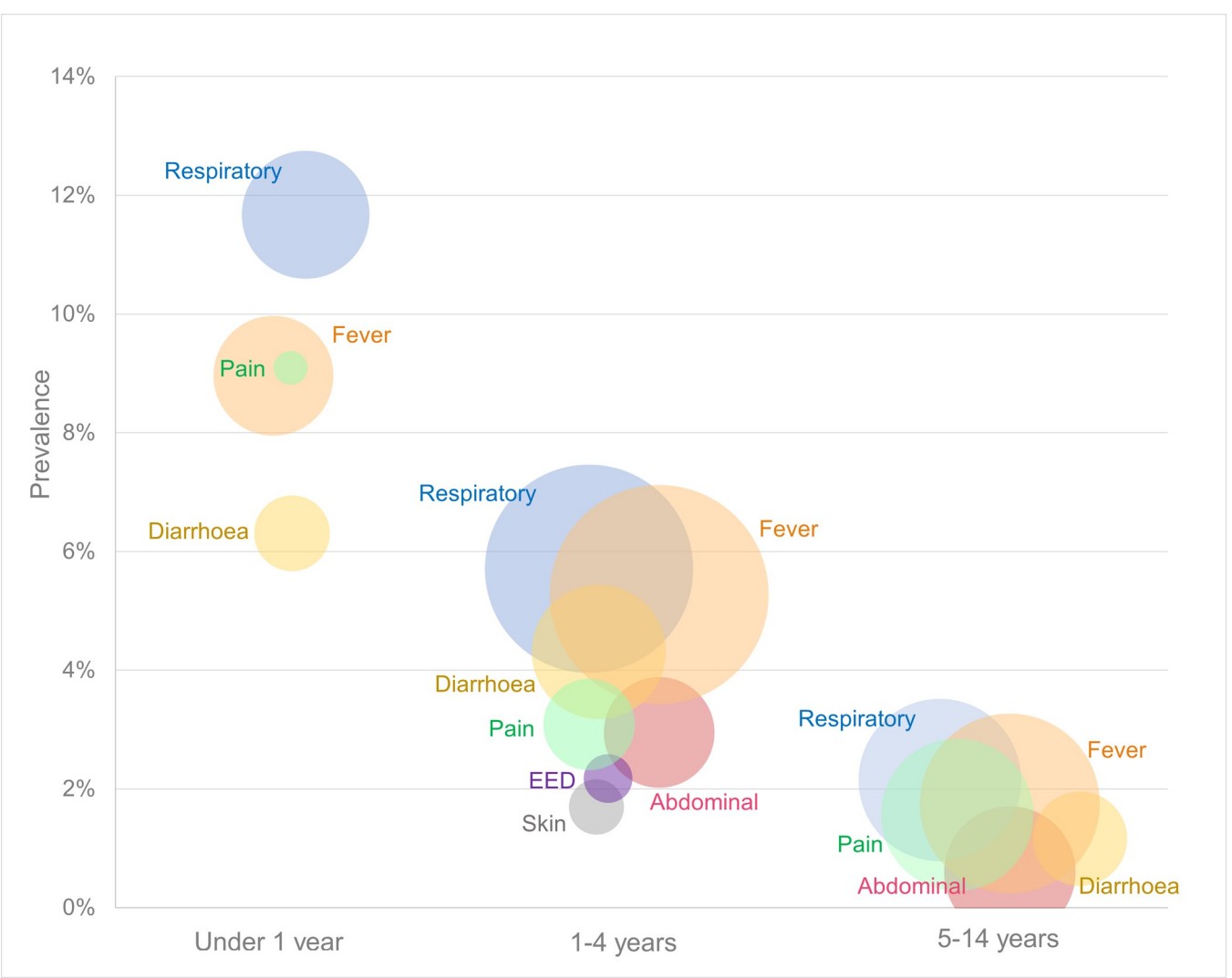

**Fig 2. Prevalence of severe or moderate hypoxaemia (SpO$_2$<94%) among acutely unwell children and adolescents presenting to HCIII facilities in Uganda, Feb-Apr 2021, by presenting complaint and age.** Notes: Size of bubble represents the number of participants, with hypoxaemia prevalence increasing up the y axis. Categories are not mutually exclusive, with many participants reporting multiple presenting complaints, however the bubble overlap is not necessarily proportional to actual presenting complaint overlap. Presenting complaint categories with less than 5 cases were excluded. EED = eye, ear, or dental complaint.

Oxygen was available in very few facilities and used for only 3 patients (0.06%) during the study period. A severely hypoxaemic (SpO$_2$ 61%) infant with febrile respiratory illness was the only patient with SpO$_2$<90% who received oxygen therapy (1/27, 3.7%). Two other patients with SpO$_2$ 93% and 98% were recorded as receiving oxygen.

### Follow up of children with hypoxaemia: Hospital attendance, admission, outcomes

**Hypoxaemia SpO$_2$<93%.**   We followed up 87 children aged under 15 years with SpO$_2$<93% and successfully obtained referral and outcome data for 61 (70%) children (including 56 under 5 years of age) (**Fig 3**). Of those with complete data, 6 (10%) had been referred to

a larger facility at the initial clinic visit, 10 (16%) had sought care at another facility, 6 (10%) were admitted overnight, and 44 (72%) had fully recovered at day 7 follow up. Of the 51 who did not attend another facility, the most commonly cited reason was that the caregiver thought it was unnecessary (42, 82%), with other reasons including cost (8, 16%), and distance or lack of transport (3, 6%).

Most children who had "not fully recovered" by day 7 had neither been referred by HCW nor sought care elsewhere (12/17, 71%). The only recorded death was of a severely hypoxaemic infant with pneumonia who was treated with appropriate antibiotics and referred (but not given oxygen) who died soon after arriving at the referral facility.

**Severe hypoxaemia SpO$_2$<90%.**   Despite study staff recommending referral for all 24 children with severe hypoxaemia (SpO$_2$<90%) a minority (10, 42%) were referred by the treating healthcare worker (**S3 Fig**). Healthcare workers gave various reasons for not referring including: they felt confident managing the child locally (4, 29%); caregiver declined referral or was worried about cost (3, 21%); patient had already left facility (2, 14%); unknown (5, 38%). While overnight admission for children was not routinely provided at participating facilities, at least four severely hypoxaemic children were observed for a period before being discharged home, including one who received oxygen therapy. Of severely hypoxaemic children who were referred, all (6/6) attended another facility, 67% (4/6) were admitted (1 died prior to admission), and half (3/6) had fully recovered on day 7. Of severely hypoxaemic children who were not referred, 20% (2/10) subsequently sought care, 10% (1/10) were admitted, and 90% (9/10) had fully recovered by day 7.

## Discussion

To our knowledge this is the first study reporting hypoxaemia prevalence among children, adolescents, and adults presenting to primary care facilities, providing unique data on hypoxaemia prevalence, referral patterns, and outcomes.

Our findings suggest that, while overall prevalence of hypoxaemia in Ugandan primary care facilities is low, young children (U5) are a key risk group for whom routine pulse oximetry assessment may improve identification and management of hypoxaemia. Current referral practices are missing a significant number of severely unwell children with hypoxaemia and pulse oximetry may be a useful tool in helping HCWs prioritise patients and facilitate referral.

### Hypoxaemia in children

**All presentations.**   We found that moderate (SpO$_2$ 90–93%) or severe hypoxaemia (SpO$_2$<90%) was relatively common among children U5 presenting to primary care facilities (1.3%, 95% CI 0.9–2.1, had SpO$_2$<90% and 4.9%, 3.9–6.1, had SpO$_2$ 90–93%) but uncommon among older children aged 5–14 years (0.3%, 0.1–1.0, SpO$_2$<90%, 1.1%, 0.6–2.0, SpO$_2$ 90–93%). Assuming similar prevalence nationally and throughout the year, this equates to >40,000 children U5 presenting to HCIIIs in Uganda with SpO$_2$<94% annually, including ~8,700 with severe hypoxaemia (SpO$_2$<90%)–of whom <50% are currently identified, treated, or referred.

A single-site study in Papua New Guinea reported that 1.4% (23/1663) of acutely unwell children U5 presenting to a rural clinic had SpO$_2$ ≤90% and an additional 2.4% (40/1663) had SpO$_2$ 91–93% [27]. Similar estimates were reported from interim data from primary care facilities in Malawi (**Table 3**) [35].

**Pneumonia.**   We found that 1.6% (95% CI 1.0–2.5) of children U5 presenting to participating Ugandan health facilities with respiratory complaints had SpO$_2$<90% and an additional 5.8% (4.6–7.3) had SpO$_2$ 90–93%. When restricted to those with specific diagnosis of

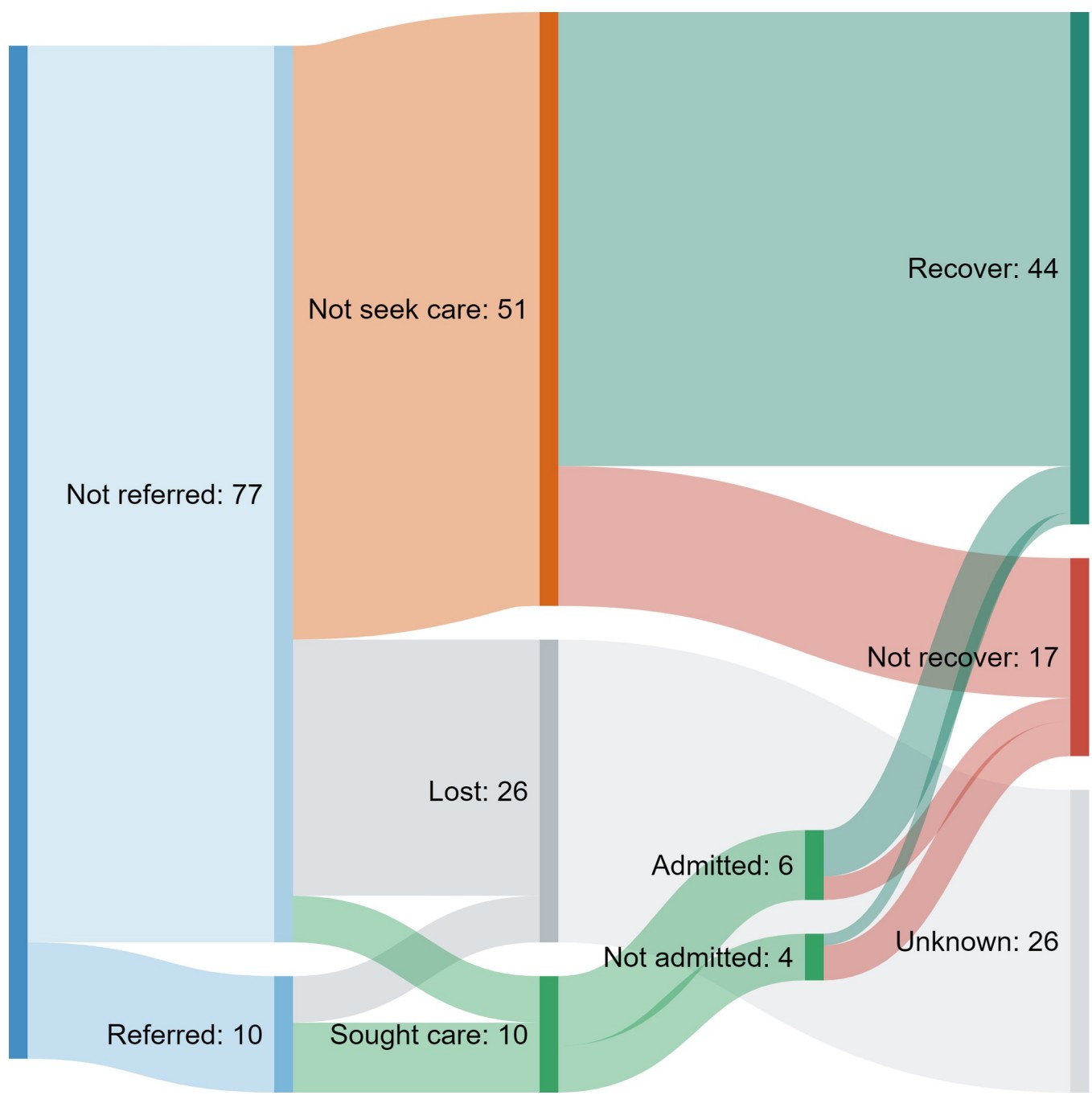

**Fig 3. Flow chart showing referral, facility attendance, and day 7 outcomes for 87 children with hypoxaemia SpO2<93%.** Numbers correspond to the underlying panel and adjacent coloured bar (right) depicting how many participants from one category progressed to the next category. Individual participant care pathways are not depicted.

pneumonia or severe pneumonia, 5.3% (2.5–10.7) had $SpO_2<90\%$ and an additional 13.5% (8.7–20.5) had $SpO_2$ 90–93%. However, the relatively low pneumonia diagnosis case numbers may suggest significant under-diagnosis of WHO-defined pneumonia.

A systematic review of hypoxaemia prevalence reported that around 13% of children aged under 5 years with severe or very severe pneumonia had hypoxaemia (typically defined as

**Table 3. Comparison of U5 hypoxaemia prevalence findings with other studies in primary care settings.**

| Study population | N | Severe hypoxaemia SpO$_2$<90% | Moderate hypoxaemia SpO$_2$ 90–93% |
|---|---|---|---|
| **Children U5 –all comers** | | | |
| *Uganda HCIII* | *1551* | *1.3% (0.9–2.1)* | *4.9% (3.9–6.1)* |
| Malawi (2020) [35]–Mchinji district | ~3000 | 0.6% | 5.4% |
| Papua New Guinea (2019) [27]—rural | 1663 | 1.4%* | 2.4%* |
| **Children U5 –pneumonia / respiratory illness** | | | |
| *Uganda HCIII–"respiratory complaints"* | *1151* | *1.6% (1.0–2.5)* | *5.8% (4.6–7.3)* |
| *Uganda HCIII–"pneumonia diagnosis"* | *133* | *5.3% (2.5–10.7)* | *13.5% (8.7–20.5)* |
| Nigeria (2021) [35]–"pneumonia or severe pneumonia", urban. | 870 | 7.5% | 10.2% |
| Bangladesh (2021) [35]–"pneumonia or severe pneumonia" | 9619 | 1.9% | 7.0% |
| Ethiopia (2020) [11]–"pneumonia or severe pneumonia" | 928 | 14.5% | - |
| Malawi (2016) [14]–"pneumonia", rural. | 13266 | 4.9% | 8.8% |
| The Gambia (1991) [26]–"pneumonia", rural. | 1033 | 10.2% | - |

95% confidence intervals only available from this current Uganda HCIII study. Data presented at Child Health Taskforce meeting [35] was confirmed by personal correspondence with Helena Hildenwall (Malawi), Carina King (Nigeria), Eric McCollum (Bangladesh), August/September 2021.

*The 2019 Papua New Guinea study reported SpO$_2$≤90%.

SpO$_2$<90%) [3]. However, most included studies were in hospital settings. A single primary care study from the early 1990s found 10.2% (105/1033) of children with pneumonia presenting to health centres or mobile clinics in the Gambia had SpO$_2$<90% (**Table 3**) [26]. More recent data from Malawi reported that 4.9% (652/13266) of children U5 with pneumonia presenting to community or village clinics had SpO$_2$<90% and an additional 8.8% (1170/13266) had SpO$_2$ 90–92% [14], with similar estimates from interim data from rural Bangladesh and urban Nigeria (**Table 3**) [35].

Differences in these hypoxaemia estimates are likely influenced by variance in pneumonia case definition and aetiology, contextual factors such as access to and thresholds for seeking care, prevalence of malnutrition and other risk factors for severe disease, and level of service provided at the participating primary care facility.

**Referral patterns.** We observed relatively low rate of referral for young children with hypoxaemia even though study staff recommended referral to the treating HCW and offered referral support. Previous studies have shown a markedly increased risk of death among children who present with hypoxaemia [1,2,36–38], including in primary care settings. This risk of death is proportional to the degree of hypoxaemia with higher risk of death even among children with moderate hypoxaemia (SpO$_2$ 90–93%) [1,2,36–39].

Referral decision-making is a complex process involving HCW and caregiver judgement of how unwell the child is and the cost-benefit of going to hospital and decisions are influenced by sociocultural and structural factors [40]. This was reflected in the feedback from HCWs and caregivers, for whom multiple barriers to referral existed even with the offer of referral assistance.

Importantly, treating HCWs had not received training on pulse oximetry or hypoxaemia, most had never seen a pulse oximeter, and hypoxaemia was not part of their standard IMCI guidelines. Previous studies suggest that if these HCWs were provided with equipment, training, and modification of their treatment guidelines they could quickly become skilled at using oximetry and using it to improve decision-making [1,10,14,15,41]. Similarly, caregivers in our study were not familiar with oximetry and it was not used to explain or persuade. Hospital experience from Nigeria suggests that caregivers respond positively to oximetry and may be

more likely to follow HCW recommendations if hypoxaemia is explained [24]. Nonetheless, barriers to hospital referral are many and the introduction of pulse oximetry will not fix all the referral problems. This may be an argument to not only introduce pulse oximetry but to also make oxygen and admission services available more locally.

## Hypoxaemia in adolescents and adults

The COVID-19 pandemic has highlighted the risk of hypoxaemia for adults and stimulated the introduction of pulse oximetry into hospitals, primary health and community care in many countries [42–44]. However we have not seen any data on hypoxaemia among adults presenting to primary care in LMICs for either respiratory or non-respiratory conditions.

We found that hypoxaemia is uncommon among acutely unwell adults presenting to primary care facilities in Uganda (0.1% had $SpO_2<90\%$ and 0.5% had $SpO_2$ 90–93%). However, hypoxaemia was more common among adolescents and adults with respiratory complaints (0.2% had $SpO_2<90\%$ and 0.6% had $SpO_2$ 90–93%) and older adults 50+ years (0.3% had $SpO_2<90\%$ and 1.2% had $SpO_2$ 90–93%).

## Implications for referral and treatment guidelines

Current WHO primary care guidelines recommend pulse oximetry "if available" for children U5 with respiratory complaints and recommend admission and oxygen therapy for those with $SpO_2<90\%$ [9,34,45]. Existing guidelines recommend targeting of higher saturation levels ($SpO_2>94\%$) for those with particular risk (e.g. severe anaemia, heart failure, acute neurological condition) [33,34], and recognise that lower saturations may be better tolerated by people living at high-altitude who have physiologically adapted to low oxygen environments [46].

However, saturation thresholds for administering oxygen therapy are different to thresholds for clinical concern and guidelines developed for hospital settings do not necessarily translate directly to primary care settings. In view of the limited data about hypoxaemia and pulse oximetry in primary care settings we cannot make strong recommendations regarding pulse oximetry in primary care. However, we have identified four key areas that warrant consideration and further exploration.

First, **routine pulse oximetry is likely to detect severe illness in children and adults presenting to primary care that would otherwise go unrecognised**. While hypoxaemia is challenging to detect using clinical features across all age groups [1,8,47–49], the high prevalence of hypoxaemia in children U5 highlights this as a priority cohort for routine pulse oximetry screening in primary care. Current IMCI guidelines suggest pulse oximetry for children U5 presenting with respiratory complaints [9]. Our data suggests that this approach would require screening of three-quarters of children and would miss 10% of children with severe hypoxaemia. A simpler approach may be to screen all acutely unwell children U5 and encourage oximetry use for older children and adults with signs of respiratory disease or severe illness.

Second, **adoption and sustained use of pulse oximetry relies on HCWs being convinced that it provides a benefit** [24]. Perceived benefit is influenced by prevalence of hypoxaemia (how frequently they will detect an abnormality that is clinically relevant) but also by whether hypoxaemia detection leads to practice and outcome change. Our study strengthens evidence showing that the inclusion of hypoxaemia in referral guidelines could change the number of children U5 identified for referral [14]. However, whether this translates to actual change in referral practices and behaviours needs further evaluation.

Third, **the hypoxaemia threshold for clinical concern and referral should probably be more inclusive than the threshold for providing oxygen therapy**. In the absence of specific data, current WHO primary care guidelines for referral use the same threshold as general

hospital guidelines for administering oxygen therapy (SpO$_2$<90%) [9,34]. However, the clear association between moderate hypoxaemia and poor clinical outcomes suggests that even moderate hypoxaemia (SpO$_2$ 90–93%) should be an indication for strong clinical concern and consideration for referral and admission for period of observation–particularly in the presence of comorbid conditions–or at least prompt more frequent review. Previous studies have found that cut-points of <92% or <94% provide better discrimination for severe disease (and poor outcome) than <90%, but such decisions for guidelines must also consider the local health system and individual clinical context and economic implications from a societal perspective [2,27,50].

Fourth, **while HCWs can quickly learn to use pulse oximeters effectively, practical challenges exist in translating this into sustained performance**. Pulse oximetry is a simple practice but we and others have seen variability in the proportion of failed or implausible readings as new users develop competence [14,41,51]. HCWs need to build confidence and skill in performing oximetry and interpreting results. Performance could be negatively impacted by infrequent practice, lack of perceived benefit (e.g. unable to treat or refer even after detection), lack of technical support resulting in repeated failure, high frequency of false positives reducing trust, lack of authoritative guidance in policy and procedures, negative opinions or critique from supervisors [24].

## Limitations

Our study involved 30 primary health care facilities across two regions in Uganda providing a representative sample and employed prospective data collection methods to maximise the quality of data. We had very high consent and completion rates, and very high success rates for pulse oximetry. However, over 1000 children attended without an adult caregiver and could therefore not give consent. This may have introduced selection bias with under-representation of older and less severely unwell children (who may be more likely to attend without an adult caregiver). We relied on HCW diagnostic classifications and did not collect independent data to validate diagnoses or distinguish between different respiratory illnesses (e.g. pneumonia, bronchiolitis, asthma, chronic obstructive pulmonary disease).

We had high rates of drop-out (30%) from post-visit follow up despite multiple attempts to contact consented families by phone. Sensitivity analysis showed no difference in reported care-seeking or clinical status on day 7 between those who were contacted on first, second, or third attempt, suggesting non-response was random. But it is possible that those who were not able to be contacted differed in risk (e.g. socioeconomic status) or outcome profiles and follow-up data should be interpreted with caution.

To obtain data from many facilities we restricted the period of data collection to 4 weeks at each facility, with data collectors moving between facilities. Data collection occurred over the warm, dry season months that are typically low-incidence for ARI and malaria and may underestimate the case numbers and hypoxaemia prevalence in other seasons [52]. At the time of this study (February to April 2021), Uganda was experiencing the COVID-19 pandemic, with data collection commencing shortly after a second major wave of infections had subsided and restrictions loosened in January 2021. While we observed similar service activity compared to previous years, changes in care-seeking, hygiene and social behaviour may have influenced participant characteristics and reduced the prevalence of ARI.

Pulse oximeter accuracy depends on the quality of the oximeter probe, internal computer/algorithm, and how it is used [53]. In laboratory conditions, pulse oximeters provide oxygen saturations with accuracy of +/- 2–3% and can be less accurate in patients with dark skin, low perfusion states, very low saturations, and those who are moving (e.g. crying child) [43,54].

Previous studies have raised concerns about the quality of some low-cost oximeters, particularly finger-tip oximeters [53–56]. We used handheld oximeters that met global quality standards and did not conduct any independent validation of results.

### Interpretation and generalisability

Our study represents hypoxaemia prevalence and practices among primary care (HCIII) facilities in two regions of Uganda. While the specific case-mix and referral findings must be understood in context, the broader findings regarding hypoxaemia burden and role for pulse oximetry are generalisable to primary care settings across the region. In the context of increasing pulse oximeter affordability, availability, and HCW acceptance, data from our observation study can inform policy, programming, and future implementation trials. Important future research questions should assess: effects of pulse oximetry on practice change and clinical outcomes; alternate cut-points for decision-making (e.g. $SpO_2$ <94% versus <90%); alternate treatment models for those with hypoxaemia (e.g. referral/admission versus day-care [57]); time and resource cost: benefits from both health service and societal perspectives; and contextual influences on adoption and impact.

### Conclusions

Hypoxaemia is common among acutely unwell children U5 presenting to Ugandan primary care facilities, and routine pulse oximetry assessment has the potential to improve referral, management and clinical outcomes. Effectiveness, acceptability, and feasibility of pulse oximetry for primary care should be investigated in implementation trials, including economic analysis from health system and societal perspectives.

### Supporting information

**S1 Fig. Map of Uganda with study regions/districts highlighted.** Blue shaded areas indicating the approximate catchment areas of Mubende and Jinja Regional referral hospitals.
(TIF)

**S2 Fig. Bimodal age distribution of hypoxaemia ($SpO_2$<94%) among acutely unwell children, adolescents, and adults presenting to HCIII facilities in Uganda, Feb-Apr 2021.** Number labels on each bar represent the number of hypoxaemia cases.
(TIF)

**S1 Table. Demographic and clinical features of 5780 acutely unwell children, adolescents, and adults presenting to primary care (HCIII) facilities in Uganda, Feb-Apr 2021.** Presenting complaints as reported by patient/caregiver. Diagnosis as recorded by treating healthcare worker. P-values obtained from Pearson's chi-squared or Fishers exact test as applicable. ARI = acute respiratory infection; STI–sexually transmitted infection; UTI–urinary tract infection.
(DOCX)

**S2 Table. Prevalence of hypoxaemia among acutely unwell children, adolescents and adults presenting to HCIII facilities in Uganda, Feb-Apr 2021 (extended results).** ARI = acute respiratory infection; CI = confidence interval; Dx = diagnoses; GIT = gastrointestinal tract; STI = sexually transmitted infection; UTI = urinary tract infection.
(DOCX)

**S3 Table. Predictors of hypoxaemia among children, adolescents, and adults presenting to HCIII facilities in Uganda, using mixed-effects logistic regression.** Final model = after

backward stepwise selection using P<0.05 as the primary determinant for exclusion at each step; CI = confidence interval; ICC–inter-cluster correlation coefficient.
(DOCX)

**S4 Table. Predictors of referral among children, adolescents, and adults presenting to HCIII facilities in Uganda, using mixed-effects logistic regression.** Final model = after backward stepwise selection using P<0.05 as the primary determinant for exclusion at each step; CI = confidence interval; ICC–inter-cluster correlation coefficient.
(DOCX)

## Acknowledgments

The authors are grateful to the field supervisors (Immaculate Kyalisiima and Joan Asekenye) and enumerators (Mary Mukhaye, Anna Maria Namuyingo, Benjamin Mulinde, Kevin Athieno, Yvette Mukundwa Awino, Edwin Bbosa, Florence Acen, Noel Aciro, Leenah Aheebwe, Hadijah Ajilong, Christine Karungi, Sharon Nakamanya, Racheal Nalika, Lilian Nazibanja, Veronica Nambi Ssebunya) who conducted the field work with diligence and dedication.

## Author Contributions

**Conceptualization:** Hamish R. Graham, Yewande Kamuntu, Jasmine Miller, Felix Lam, Freddy Eric Kitutu.

**Data curation:** Jasmine Miller, Blasio Kunihira, Felix Lam.

**Formal analysis:** Hamish R. Graham, Jasmine Miller, Anna Barrett, Felix Lam.

**Funding acquisition:** Lorraine Kabunga.

**Investigation:** Hamish R. Graham, Yewande Kamuntu, Jasmine Miller, Blasio Kunihira, Felix Lam.

**Methodology:** Hamish R. Graham, Yewande Kamuntu, Jasmine Miller, Felix Lam, Charles Olaro, Harriet Ajilong, Freddy Eric Kitutu.

**Project administration:** Yewande Kamuntu, Jasmine Miller, Blasio Kunihira, Santa Engol, Felix Lam.

**Supervision:** Hamish R. Graham, Yewande Kamuntu, Lorraine Kabunga, Felix Lam, Charles Olaro, Harriet Ajilong, Freddy Eric Kitutu.

**Validation:** Anna Barrett, Blasio Kunihira, Santa Engol.

**Visualization:** Hamish R. Graham, Anna Barrett.

**Writing – original draft:** Hamish R. Graham.

**Writing – review & editing:** Hamish R. Graham, Yewande Kamuntu, Jasmine Miller, Anna Barrett, Felix Lam, Charles Olaro, Harriet Ajilong, Freddy Eric Kitutu.

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
