## [Decision Letter · Decision Letter 0]

31 Jan 2022

PGPH-D-21-01118

Hypoxaemia prevalence and management among children and adults presenting to primary care facilities in Uganda: a prospective cohort study.

Dear Dr. Graham,

Thank you for submitting your manuscript to PLOS Global Public Health. After careful consideration, we feel that it has merit but does not fully meet PLOS Global Public Health’s publication criteria as it currently stands. Therefore, we invite you to submit a revised version of the manuscript that addresses the points raised during the review process.

We look forward to receiving your revised manuscript.

Kind regards,

Benjamin Tsofa, DrPH

Section Editor

Journal Requirements:

1. Please include a complete copy of PLOS’ questionnaire on inclusivity in global research in your revised manuscript. Our policy for research in this area aims to improve transparency in the reporting of research performed outside of researchers’ own country or community. The policy applies to researchers who have travelled to a different country to conduct research, research with Indigenous populations or their lands, and research on cultural artefacts. The questionnaire can also be requested at the journal’s discretion for any other submissions, even if these conditions are not met.  Please find more information on the policy and a link to download a blank copy of the questionnaire here: https://journals.plos.org/plosone/s/best-practices-in-research-reporting. Please upload a completed version of your questionnaire as Supporting Information when you resubmit your manuscript.

2. Please provide separate figure files in .tif or .eps format only, and remove any figures embedded in your manuscript file.

3. We noticed that you used "unpublished data" in the manuscript. We do not allow these references, as the PLOS data access policy requires that all data be either published with the manuscript or made available in a publicly accessible database. Please either remove these references, or amend the supplementary material to include the referenced data.

4. Please update the completed 'Competing Interests' statement, including any COIs declared by your co-authors. If you have no competing interests to declare, please state "The authors have declared that no competing interests exist". Otherwise please declare all competing interests beginning with the statement "I have read the journal's policy and the authors of this manuscript have the following competing interests:"

5. We have noticed that you have uploaded supporting information but you have not included a list of legends.  Please add a full list of legends for all supporting information files (including figures, table and data files) after the references list. 

6. Please amend your detailed Financial Disclosure statement. This is published with the article, therefore should be completed in full sentences and contain the exact wording you wish to be published.

ii). State the initials, alongside each funding source, of each author to receive each grant.

iii). State what role the funders took in the study. If the funders had no role in your study, please state: “The funders had no role in study design, data collection and analysis, decision to publish, or preparation of the manuscript.”

Additional Editor Comments (if provided):

Reviewers' comments:

Reviewer's Responses to Questions

**Comments to the Author**

1. Does this manuscript meet PLOS Global Public Health’s publication criteria? Is the manuscript technically sound, and do the data support the conclusions? The manuscript must describe methodologically and ethically rigorous research with conclusions that are appropriately drawn based on the data presented.

Reviewer #1: Yes

Reviewer #2: No

2. Has the statistical analysis been performed appropriately and rigorously?

Reviewer #1: Yes

Reviewer #2: Yes

3. Have the authors made all data underlying the findings in their manuscript fully available (please refer to the Data Availability Statement at the start of the manuscript PDF file)?

Reviewer #1: Yes

Reviewer #2: Yes

4. Is the manuscript presented in an intelligible fashion and written in standard English?

Reviewer #1: Yes

Reviewer #2: Yes

5. Review Comments to the Author

Reviewer #1: Abstract:

• Intro and Para 1 of main text Introduction – It is stated that: ‘Low blood oxygen levels (hypoxaemia) are common among children and adults admitted to hospital 30 and it increases their risk of death 5-fold and thus requires prompt detection and treatment.’ – I think the data indicate that hypoxaemia is associated with a 5 fold increase in the odds of death; stating causality in such strong terms would suggest the risk of death can be reduced 5 fold by correcting hypoxaemia which is not as far as I am aware proven.

• Secondary outcome in the abstract is stated as “severe/moderate hypoxaemia (SpO2 90-93%) by 13 age/sex/complaint” – this wording is confusing, is this all Sp02<94%?

• Conclusion – the recommendations include “Effectiveness, acceptability, and feasibility of pulse oximetry 26 and oxygen therapy for primary care should be investigated in implementation trials.” – Do the authors think that costs and consequences to families and health systems should also be assessed as there may well be implications from referrals that do not improve clinical outcomes?

General comments:

• No mention is made of whether the sites of this study are equipped to manage asthma in any way if they feel this diagnosis is present?

• Throughout the paper it is sometimes hard to keep track of what is meant by hypoxaemia, sometimes the term is used alone, sometimes qualified by either severe or moderate. In general it might be better to use the pulse oximetry values throughout rather then the term hypoxaemia even if qualified by an adjective. In a similar vein it would be helpful to carefully use <94% to mean all those with a saturation lower than this or 90-93% for the specific group wit those values (see Fig3 title for example).

• In the discussion section where results are repeated as % values please continue to indicate the 95% CI around the estimates.

• The section in the discussion on hypoxaemia in children repeats much of what has been stated in the results and what is provided in Table 3 is stated in the narrative text too. This section of the discussion could be made more concise and clearly state what added points are being made.

Specific comments

o Line 303 – “We found that moderate or severe hypoxaemia was relatively common among children U5 presenting to primary care facilities (1.3% had SpO2<90% and 4.9% had SpO2 90-93%) but uncommon among older children aged 5-14 years (0.3% SpO2<90%, 1.4% SpO2 90-93%). Assuming similar prevalence nationally and throughout the year, this equates to >40,000 children U5 presenting to HCIIIs in Uganda with hypoxaemia annually (including ~8,700 with severe hypoxaemia) – most of whom are not currently identified, treated, or referred.” - While the findings restated here are those found in the text the implicit messaging here is that those with oxygen saturation of 90 to 93% should be ‘identified, treated and referred’. Is this the author’s intention and if so what evidence supports this course of action? This might result in a 300% increase in respiratory referrals? Given recent findings on the potential for error in dark skins this challenge seems worth a mention here.

• Line 339 – the 1st reference is given to support this statement ‘This risk of death is proportional to the degree of hypoxaemia with higher risk of death even among children with moderate hypoxaemia (SpO2 90-93%)’ but the original reference does not provide a clear articulation of this apparent risk and is from an inpatient population (including neonates) with much higher rates of hypoxaemia overall and much higher risks of death. Please clarify / correct the statement or remove it.

• Line 345 – “Our study was not powered to detect differences in mortality, but it is concerning that a significant proportion of children who were not referred were subsequently taken to a referral facility by caregivers, required admission, and had not fully recovered by day 7” – it is probably helpful not to use the term ‘significant’ here – please state the actual numbers meant and perhaps caveat this statement with the fact that none of these children as far as I can tell died while it seems some of those referred or admitted had also not recovered? It is also worth remembering that there are real costs borne by families from referral that are both direct and indirect (eg loss of income). In essence pulse oximetry in primary care is being proposed as a screening tool and therefore we have to carefully consider the consequences and costs of both false positives and false negatives especially in situations where the true prevalence can be very low. Given that the readings taken in this study were done by supernumerary, trained staff who took up to 3 readings it is hard to know what these false positive and false negative % would be in ‘real life’ and what effects of routine oximetry would have on working practices. (These issues are relevant to later discussions on generalisability)

• Line 388 – this statement is made – “Second, adoption and sustained use of pulse oximetry relies on HCWs being convinced that it provides a benefit”. – Would the authors agree that to recommend adoption and sustained use of pulse oximetry we first need to provide good evidence of its benefit before asking health workers to take it on?

• Line 395 – “Third, the hypoxaemia threshold for clinical concern and referral should probably be more inclusive than the threshold for providing oxygen therapy.” – Would the authors like to comment on the variations in guidance and practice in high income countries with regard to when oxygen saturation values indicate clinical concern and how this might vary with possible disease aetiology and age (eg bronchiolitis / RSV). It is also unclear to me what is actual meant by SpO2 from 90-93% being ‘an indication for strong clinical concern and consideration for referral and admission’. Are the authors proposing this threshold replaces current guidance or not at primary care level where health workers with limited skills operate?

• Line 403 – Pulse oximetry may appear to be a simple practice but what are the challenges in providing and sustaining its use across nearly 1000 HCIII in Uganda – how has progress been in providing this technology in HCIV or district general hospitals in Uganda?

Minor

• Line 198 – is ‘respiratory’ a presenting complaint? Seems words missing?

• Line 227 - Hypoxaemia prevalence was most common among children diagnosed with ARI (8.1%, 6.4 to 10.2), malaria (5.3%, 4.0 to 7.1), or sepsis (17.1%, 8.2 to 32.3), and particularly high for those diagnosed with pneumonia (18.8%, 13.0 to 26.4) – this is a little hard to interpret since the diagnosis was made with knowledge of the pulse oximetry value?

• I am not sure this ‘result’ has much meaning - Oxygen was rarely available and was provided to only 3 patients (0.06%). A severely hypoxaemic (SpO2 61%) infant with febrile respiratory illness was the only patient with SpO2<90% who received oxygen therapy (1/27, 3.7%). Two other patients with SpO2 93% and 98% were recorded as receiving oxygen – is this because only one place had oxygen for only some of the time or all places had oxygen rarely but didn’t use it when they could?

• Line 269 – were the 6 admitted overnight the same as those who were referred to a larger facility or who went elsewhere for care?

• Line 298 - Current referral practices are missing a significant number of severely unwell children with hypoxaemia and pulse oximetry may be a useful tool in helping HCWs prioritise patients and facilitate referral. – However, the majority of those with severe hypoxaemia but not referred recovered without referral care?

• Line 320 - However, the relatively low pneumonia diagnosis case numbers suggest significant under-diagnosis. – What is the justification for this statement?

• Line 33 & 332 – presumably variability in the aetiology of respiratory infection and/or asthma also influences apparent rates of hypoxaemia?

Reviewer #2: This study is unique because it explores hypoxaemia in primary care and includes both adults and children. Most studies in low income countries to date have studied hypoxaemia in inpatients - at secondary and tertiary referral - with a focus either on children or adults. I feel that this study will be a useful addition to the existing literature and will make an important contribution toward scaling up pulse oximetry use and promoting rational oxygen therapy in low resource settings.

Overall this was a clear and well-presented paper, reporting on the prevalence of hypoxaemia in primary care facilities in Uganda. I have the following comments which may require the authors’ response.

Abstract

1. The abstract is well written and contains pertinent information about the study.

I would suggest that the authors consider reporting the numbers of participants in the study in the abstract because at present a reader will have no idea about the size of this study by reading through the abstract alone.

2. The methods in the abstract mentions that blood oxygen level was assessed but does not indicate how this assessment was done – it will be helpful to mention pulse oximetry in the abstract methods because the conclusion of the study is on pulse oximetry yet there is no mention of pulse oximetry before this conclusion.

3. Information on follow up is pertinent in a cohort study and it is important for the authors to provide some detail (at least for the abstract an indication of period of follow up would be useful).

Introduction

4. The introduction provides a clear scientific background and rationale for the study, and the objectives are stated.

5. The authors state that they are aware of only three studies that report hypoxemia prevalence in primary care settings. However, they proceed to provide references for two studies (Reference 11 from Ethiopia and 14 from Malawi). Did they forget to reference the study from The Gambia?

6. I also conducted a search and found the study by Blanc et al on pulse oximetry in young children with severe illness in outpatient clinics in Papua New Guinea (https://doi.org/10.1371/journal.pone.0213937). It would be good to get the authors’ perspective on why this study and possibly others were not considered as relevant – this may require the authors to qualify their statement as to why they feel only the three studies they identified address the issue at hand. For example, in the discussion you suggest that there are no published studies on hypoxemia prevalence among general populations presenting to outpatient. I suggest that such statement in the introduction and discussion are made consistent and while avoiding broad, unqualified statements.

Methods

7. The study design in presented as a prospective cohort study. My appraisal of the study design is that the prospective cohort design applies to only 87 children with severe and moderate hypoxaemia (line 264 – 291), out of a total of 5813 participants in the study. Most participants contribute only cross-sectional data used to determine prevalence and predictors of hypoxaemia. This detail is obscured in the title and methods and I feel that this may have an important bearing on the classification of the study design. The authors may want to give a clearer description of the study design highlighting these issues.

8. For a prospective cohort study I feel that the authors did not provide pertinent details that are required for a cohort design. The information on the follow up period is very limited, and is mentioned only in one sentence in data collection with respect to consenting for data collection (line 139 i.e. … patients/caregivers were also invited to consent to a follow up phone call after 7 days…). In prospective cohorts follow up is planned as part of the design and should be well described within the study design and or participants’ sections.

9. The authors provide a detailed description of the study setting including levels of health facilities in Uganda and typical health services provided. A potentially relevant aspect of study settings in hypoxaemia studies is altitude because of its association with oxygen levels in blood. The authors could consider providing details of the altitude of study sites.

Selection of health facilities is described in detail.

10. The sample size calculation assumed that 3500 children under 5 would be available in the study but the study ended up recruiting about half of this number. What is the implication of this sample size that was obtained after recruitment?

11. You use two definitions of severe hypoxemia in different sections of the manuscript i.e. SpO2 < 93 and SpO2 < 94. Please check these definitions that appear to be inconsistent.

12. The multilevel random effect model is appropriate for the type of data in this analysis. The description of the model building is not adequate because the author simply states they used backward stepwise selection – additional details on the variable selection (how you determined the least significant variables at each step) and the stopping rule will be useful.

Results

13. A significant number of exclusion is in children who were unaccompanied during the visit to facilities. It will be helpful to comment on this group as it is a possible source of selection bias.

Some of the data that you present in text in the results section do not appear in the tables, for example on the prevalence of hypoxemia you report that “Overall hypoxaemia (SpO2<94%) prevalence among children and adults presenting with acute illness to HCIII primary care facilities in Uganda was 2.2% (95% CI 1.9 to 2.7) (Table 2).” However, these overall results do not appear in table 2. This is the case for several other results sections where you report overall results in text. Please consider adding a column for the overall results in Table2.

14. Considering the percentages in Table 2, I am not completely in agreement that hypoxemia showed a bimodal distribution with age.

15. The authors should consider briefly describing the WHO IMCI guidelines for performing pulse oximetry so as to provide context for readers who are not familiar with the guidelines to interpret the findings in this study.

16. The analysis of pathway of care was presented quite effectively using Figure 3.

Discussion

17. The authors provide a balanced interpretation of the results, including its generalisability. In terms of limitations it would be useful to comment on the potential direction and magnitude of the potential bias that could result from high rates of drop-outs identified in the study limitations

6. PLOS authors have the option to publish the peer review history of their article (what does this mean?). If published, this will include your full peer review and any attached files.

**Do you want your identity to be public for this peer review?** For information about this choice, including consent withdrawal, please see our Privacy Policy.

Reviewer #1: No

Reviewer #2: No

---

## [Decision Letter · Decision Letter 1]

15 Mar 2022

Hypoxaemia prevalence and management among children and adults presenting to primary care facilities in Uganda: a prospective cohort study.

PGPH-D-21-01118R1

Dear Graham,

We are pleased to inform you that your manuscript 'Hypoxaemia prevalence and management among children and adults presenting to primary care facilities in Uganda: a prospective cohort study.' has been provisionally accepted for publication in PLOS Global Public Health.

Best regards,

Benjamin Tsofa, DrPH

Section Editor

Reviewer Comments (if any, and for reference):

Reviewer's Responses to Questions

**Comments to the Author**

1. If the authors have adequately addressed your comments raised in a previous round of review and you feel that this manuscript is now acceptable for publication, you may indicate that here to bypass the “Comments to the Author” section, enter your conflict of interest statement in the “Confidential to Editor” section, and submit your "Accept" recommendation.

Reviewer #1: All comments have been addressed

Reviewer #2: All comments have been addressed

2. Does this manuscript meet PLOS Global Public Health’s publication criteria? Is the manuscript technically sound, and do the data support the conclusions? The manuscript must describe methodologically and ethically rigorous research with conclusions that are appropriately drawn based on the data presented.

Reviewer #1: Yes

Reviewer #2: Yes

3. Has the statistical analysis been performed appropriately and rigorously?

Reviewer #1: Yes

Reviewer #2: Yes

4. Have the authors made all data underlying the findings in their manuscript fully available (please refer to the Data Availability Statement at the start of the manuscript PDF file)?

Reviewer #1: (No Response)

Reviewer #2: Yes

5. Is the manuscript presented in an intelligible fashion and written in standard English?

Reviewer #1: Yes

Reviewer #2: Yes

6. Review Comments to the Author

Reviewer #1: The comments made have been addressed. I think there is still an opportunity to point to the potential challenge of correct use and accuracy of pulse oximetry in very large numbers of facilities that have traditionally had little or no diagnostic equipment and where maintenance and procurements systems are typically very weak. What threats are there from a poorly functioning or poorly used device and how might the opportunity cost of providing such devices compare with offering clinical supervision?

Reviewer #2: (No Response)

7. PLOS authors have the option to publish the peer review history of their article (what does this mean?). If published, this will include your full peer review and any attached files.

**Do you want your identity to be public for this peer review?** For information about this choice, including consent withdrawal, please see our Privacy Policy.

Reviewer #1: No

Reviewer #2: No
